# Phase-Resolved Partial Discharge (PRPD) Pattern Recognition Using Image Processing Template Matching

**DOI:** 10.3390/s24113565

**Published:** 2024-05-31

**Authors:** Aliyu Abubakar, Christos Zachariades

**Affiliations:** Department of Electrical Engineering and Electronics, University of Liverpool, Liverpool L69 3GJ, UK; aliyu.abubakar@liverpool.ac.uk

**Keywords:** condition monitoring, electrical insulation, image processing, incipient fault, insulation defect, pattern recognition, partial discharge, PRPD, template matching

## Abstract

This paper proposes a new method for recognizing, extracting, and processing Phase-Resolved Partial Discharge (PRPD) patterns from two-dimensional plots to identify specific defect types affecting electrical equipment without human intervention while retaining the principals that make PRPD analysis an effective diagnostic technique. The proposed method does not rely on training complex deep learning algorithms which demand substantial computational resources and extensive datasets that can pose significant hurdles for the application of on-line partial discharge monitoring. Instead, the developed Cosine Cluster Net (CCNet) model, which is an image processing pipeline, can extract and process patterns from any two-dimensional PRPD plot before employing the cosine similarity function to measure the likeness of the patterns to predefined templates of known defect types. The PRPD pattern recognition capabilities of the model were tested using several manually classified PRPD images available in the existing literature. The model consistently produced similarity scores that identified the same defect type as the one from the manual classification. The successful defect type reporting from the initial trials of the CCNet model together with the speed of the identification, which typically does not exceed four seconds, indicates potential for real-time applications.

## 1. Introduction

Power system equipment constitutes invaluable assets whose reliable operation is paramount for the continuous delivery of electrical energy necessary to support the essential functions of our daily lives. Over its lifetime, it is possible for equipment to develop faults that require downtime to address. Furthermore, over time and with continuous use, equipment ages, requiring asset managers to make a choice between replacement and life extension. To prevent catastrophic failure but also to inform maintenance decisions, various condition monitoring techniques are available, but when considering the insulation of High-Voltage (HV) equipment, which can be solid, liquid, or gaseous, by far the most popular condition monitoring method is partial discharge (PD) detection. This can detect small electrical discharges that can act as precursors to more severe problems. PD activity under the influence of loading and environmental conditions can worsen over time indicating insulation degradation which, if left undiagnosed, can lead to the complete failure of equipment [1]. Hence, it is vital to detect PD activity and closely monitor its evolution over time in order to take timely and appropriate action to prevent incipient faults from escalating and avert serious repercussions.

Defects in insulation can arise from various sources. Manufacturing imperfections might introduce cavities while handling during delivery or installation might lead to mechanical damage. Operational wear, such as physical harm to the electrical apparatus, or natural aging can erode the resilience of the insulating material [2]. These defects can instigate various forms of PD, as illustrated in Figure 1. Examples include internal discharges within voids of solid or liquid insulation, surface discharges on the insulator’s external layer, corona discharges stemming from uneven electric fields at electrode tips, or treeing caused by continuous discharge effects in solid insulating materials.

The measurement of PD signals is conducted using two main approaches: on-line PD (ONPD) measurement and off-line PD (OFPD) measurement. ONPD detection is conducted while the equipment is live, requiring no de-energization or disruption of normal operation [3,4,5]; hence, the equipment is subjected to actual conditions that can affect PD activity such as HV stress, temperature, and vibration. On the other hand, OFPD detection is conducted with the equipment disconnected from the network, bringing a complete halt to normal operation [6,7,8], and requires the equipment to be energized via external means. OFPD can incur additional costs compared to ONPD because it requires the interruption of service, which can halt the operation of critical utilities that operate continuously. A variety of sensors can be used for PD detection depending on the equipment being monitored such as Ultra-High Frequency (UHF) couplers [9], High-Frequency Current Transformers (HFCTs) [10], Ultrasonics, and Transient Earth Voltage (TEV) detectors.

One of the methods used to interpret PD measurements is Phase-Resolved Partial Discharge (PRPD) pattern analysis. Each of the PD pulses captured using a PD sensor is quantified based on its phase angle $\left(\varphi \right)$ in relation to the power frequency sinusoid and its magnitude in terms of electrical charge $\left(q\right)$. The number of PD pulses $\left(n\right)$ over a pre-established time period is then visualized on a PD–charge over phase angle plot often referred to as φ−q−n or PRPD plot [11,12]. The patterns formed in PRPD plots can be used to identify specific types of insulation defects which can inform targeted maintenance operations to avoid equipment failure. However, PRPD analysis relies on the expertise of engineers with specialist knowledge, often using proprietary instruments and software since there is no standardization regarding how PRPD patterns are produced or displayed. Additionally, the method can become resource-intensive and time-consuming due to the large volume of data that needs to be continuously processed. Together with the fact that PRPD patterns are not static and can evolve over time, the application of PRPD analysis is mostly reserved for the most critical or most expensive equipment. 

Over the years, researchers have made significant efforts to come up with robust and efficient ways of analyzing PD. For example, the study in [13] introduced an automated approach for recognizing partial discharges, simulating five unique defects in transformers. These defects included a scratch on the winding insulation leading to surface discharge, a bubble in the oil, moisture in the insulation paper, a small free metal particle inside the transformer tank, and a deeply embedded sharp metal point on the tank. The approach leveraged texture features from the Gray-Level Co-occurrence Matrix (GLCM) applied to Continuous Wavelet Transform (CWT) images and utilized Principal Component Analysis (PCA) for reducing dimensionality alongside the support vector machine (SVM) for classification. In [14], grayscale PRPD images, originating from insulation flaws in transformers, were used to train the SVM algorithm with both local binary pattern and histogram-oriented gradient features. The method described in [15] introduced a combination of the Gray-Level Co-occurrence Matrix with Optimal Parameters and the support vector machine (GLCMOP-SVM). This method was designed to identify three specific transformer faults: tip discharge, air discharge, and surface discharge. The authors of Reference [16] introduced an SVM-driven PD identification algorithm. This was trained using GLCM and Tamura characteristics and attained an accuracy of 90.6%. Nonetheless, the aforementioned methods were largely based on intricate descriptive feature extraction techniques, involving complicated mathematical procedures. 

Recently, Convolutional Neural Networks (CNNs) have been embraced to improve feature extraction efficiency. The effectiveness of CNNs has been proven in various sectors like health [17,18], security [19], and business forecasting [20]. In the realm of condition monitoring, the study in [21] suggested CNNs for extracting PD features in 3D patterns from four distinct GIS defects. The authors of Reference [22] introduced a method based on CNNs for diagnosing partial discharge in power transformer equipment. The research specifically simulated six artificial insulation defects under different voltage levels. Another study used ensemble CNN models to identify PD from both recognized and unknown sources based on a set threshold [23]. The motivation behind this study was to address the limitations of using a singular feature extraction method, which might reduce accuracy if the feature is not optimally captured. While very high accuracy was reported, validation with real-world data was still pending, as noted by the authors. Nevertheless, the substantial computational requirements associated with deep learning techniques present challenges within industrial settings where computing resources may be limited. This underscores the ongoing need for research in methods for analyzing PD patterns. 

In this paper, a novel approach to PRPD pattern recognition is proposed. It employs cosine similarity within an image processing pipeline to match processed PRPD images with predefined templates that correspond to known defect patterns. Unlike other methods aiming to automate PD diagnostics, this approach is much closer to the way an expert human operator would identify PRPD patterns and determine the type of defect present in an electrical apparatus. The new method bypasses the complexity of deep learning models and focuses on leveraging the collective knowledge accumulated over decades of PD research in recording PRPD patterns and correlating them with specific insulation defects. It uses this to identify defect types directly from PRPD images quickly and efficiently. It swiftly and effectively identifies defect types directly from PRPD images. The approach is designed to be independent of the specifics of PD measurement techniques and the visual representation of PRPD patterns, allowing it to potentially integrate seamlessly with both new and existing PD monitoring systems to expedite the diagnosis of incipient insulation faults. 

## 2. PRPD Image Pre-Processing

The quality of images can deteriorate from several causes, including noise during capture or transmission or even camera-specific parameters such as quality, calibration, resolution, and lighting conditions [24]. This represents a significant hurdle in computer vision. Unwanted distortions can hamper the precision of the feature extraction process and further impact the recognition process. Therefore, it is essential to implement pre-processing methods to either reduce the imperfections or improve the quality of the dataset. The methods utilized in this study for PRPD image pre-processing are described in the following sections.

### 2.1. Noise Reduction

Noise reduction is a process aimed at enhancing the quality of an image by retaining its key features while minimizing unwanted information. A widely utilized technique for noise reduction in computer vision is Bilateral Filtering (BF). This method operates by considering both the geometric proximity and photometric similarity of pixels within a local window to determine the value of a specific pixel. In simpler terms, Bilateral Filtering calculates a weighted sum of pixels within the local window. The weights are determined based on both the intensity differences and spatial distances. This approach ensures that noise is effectively smoothed out while still maintaining the sharpness of edges. The calculation for Bilateral Filtering can be applied to any pixel located at position X, using the following (1):(1)$$I\left(X\right)=\frac{1}{C}\sum_{y\in N\left(X\right)}e^{\frac{-{\left\|y-x\right\|}^{2}}{2{\sigma_{d}}^{2}}}\cdot e^{\frac{-{\left\|I\left(y\right)-I\left(x\right)\right\|}^{2}}{2{\sigma_{r}}^{2}}}\cdot I\left(y\right)$$

The parameters $\sigma_{d}\;and\;\sigma_{r}$ in this context are utilized to control the trade-off between the spatial and intensity domains, respectively. Here, N(X) refers to a spatial local window of position X, while C stands for the normalization constant. The value of constant C is obtained by the following (2):(2)$$C=\sum_{y\in N\left(X\right)}e^{\frac{-{\left\|y-x\right\|}^{2}}{2{\sigma_{d}}^{2}}}\cdot e^{\frac{-{\left\|I\left(y\right)-I\left(x\right)\right\|}^{2}}{2{\sigma_{r}}^{2}}}$$

The values assigned to the two parameters, $\sigma_{d}\;and\;\sigma_{r}$, are instrumental in determining the final quality of the denoised output. However, the optimal values for these parameters are not easily determined by theoretical research [25]. As such, the most effective approach for defining these values often involves a process of trial and error.

### 2.2. Illumination Enhancement

Histogram equalization (HE) is a method used to improve the quality of digital images by redistributing pixel values across a specified gray range. Sometimes referred to as histogram flattening, this nonlinear stretching technique ensures that pixel values are roughly equalized within the range, leading to a more uniform distribution of gray levels. The resulting effect is an image that appears flatter and clearer. The underlying principle of HE is to distribute the gray levels evenly, which often enhances the overall contrast and clarity of the image.

### 2.3. Contrast Enhancement

To further boost the contrast of images, the Contrast Limited Adaptive Histogram Equalization (CLAHE) technique was used. This method operates by capping the contrast, thereby mitigating the impact of noise amplification. It ensures better contrast without compromising the quality of the image. β represents the limit value (clip limit), which signifies the highest permissible histogram height, and can be established through the following (3):(3)$$\beta =\frac{M}{N}\left[1+\frac{\phi }{100}\left(S_{\max}+1\right)\right]$$

N represents the grayscale value (typically set at 256), M refers to the region size, ϕ signifies the clip factor indicating the addition of a histogram with a value ranging from 1 to 100 [26], and $S_{max}$ is the maximum permissible slope. The PRPD images were sectioned into 8×8 sized areas, and the clip limit was set to 3. 

### 2.4. Image Segmentation

Image segmentation plays an important role in image processing. In this work, the k-means clustering algorithm was used to partition a PRPD image into a specific number of groups, where a region of interest was selected [27]. Using the k-means algorithm involves two distinct stages. In the first stage, the k centroid is calculated, and in the second stage, each point that has the closest centroid from the respective data point is selected. Euclidean distance is one of the common methods used to define the closest centroid. The k-means clustering is an iterative method where for each grouping, the new centroid of each cluster will be recalculated, and Euclidean distance will also be calculated between each center and each data point, and the point will be assigned to the cluster with the shortest Euclidean distance. An image with dimensions x×y can be partitioned into k number of clusters. If p(x,y) are the image pixels to be clustered, and $c_{k}$ is the center of the cluster, the steps for implementing the k-means clustering algorithm are the following:Set the number of clusters k and center.Calculate $E_{d}$ (Euclidean distance) between the center and each pixel of the input image using the following:(4)$$E_{d}=\left\|p\left(x,y\right)-c_{k}\right\|$$Assign all pixels to the closest center based on d.Then, recalculate the new position of the center using the following:(5)$$c_{k}=\frac{1}{k}\sum_{y\in c_{k}}\sum_{y\in c_{k}}p\left(x,y\right)$$Iterate until the error value is satisfied.Finally, reshape the cluster pixels into an image.

## 3. PRPD Pattern Recognition Approach

In the recent existing literature, researchers have primarily employed deep neural networks for the recognition of partial discharges. In some instances, lighter machine learning algorithms like support vector machines and k-nearest neighbors have also been used in combination. The limitation associated with these algorithms stems from the scarcity of extensive datasets required for training these resource-intensive models. While PRPD patterns can depend on the specific asset being monitored and the type of defect, when discharges are recorded and presented in this form, well-defined representations are produced. Processing these uniquely shaped patterns rather than the raw data used to form them could potentially overcome the constraints faced by deep learning algorithms.

In this study, phase-resolved plots of partial discharges in the form of images are processed by firstly isolating individual PRPD patterns, and then cosine similarity is employed to assess the degree of resemblance between each pattern and predefined pattern shape templates (Table 1). The templates are themselves created by phase-resolved plots specified in international standards [28] and the related literature, such as CIGRE brochures [29], where these are used as guidelines for the interpretation of PD measurements.

### 3.1. Cosine Similarity

Cosine similarity (*CS*) is a mathematical measure that calculates the cosine of the angle between two non-zero vectors in an inner dot product space. Cosine similarity can be computed using the following (6):(6)$$x\cdot y=\left\|x\right\|\cdot \left\|y\right\|\cdot \cos\theta$$

Here, x and y are the two given vectors, and θ is an angle between them. The similarity S is computed using the following (7):(7)$$S=\cos\theta =\frac{x\cdot y}{\left\|x\right\|\cdot \left\|y\right\|}=\frac{\sum_{i=1}^{n}x_{i}y_{i}}{\sqrt{\sum_{i=1}^{n}{x_{i}}^{2}}\sqrt{\sum_{i=1}^{n}{y_{i}}^{2}}}$$

*CS* is computed to find the shortest distance between each PRPD pattern and each of the predefined shape templates:(8)$$CS_{shortest}=\min\left[dist\left(PRPD_{pattern},Template_{image}\right)\right]$$

To measure the dissimilarity between the pattern and template, the cosine distance can be calculated which provides a score ranging from 0 to 1. A higher score indicates greater dissimilarity between the two vectors, while a lower score indicates greater similarity. The cosine similarity, which is used in this study, is calculated by subtracting the cosine distance from 1. Hence, a higher cosine similarity score indicates greater similarity between the image vectors, while a lower score suggests lower similarity.

### 3.2. Cosine Cluster Net (CCNet)

The Cosine Cluster Net (CCNet) is an image processing pipeline that includes multiple steps. A graphical representation of its workings is shown in Figure 2. It processes each two-dimensional PRPD pattern using a clustering algorithm, specifically the K-means clustering algorithm. K-means clustering is an unsupervised machine learning algorithm which is not inherently suited for feature extraction but produces cluster assignments [30]. The K-means algorithm is integrated into the pipeline as pre-processing before feeding the centroids into the cosine similarity algorithm. This allows for the segmentation of the patterns within the image from the background. The use of K-means clustering is beneficial due to its simplicity and speed, making it suitable for applications requiring real-time or near-real-time processing. It also operates efficiently with smaller datasets, offers straightforward interpretability, and is less demanding on computational resources, making it an ideal choice for scenarios where quick, clear segmentation is crucial without the complexities of deep learning. It is important to note that cosine similarity operates on one-dimensional (1D) vectors. To ensure consistency, each of the two-dimensional (2D) vectors is transformed into 1D vectors using a flattening function.

## 4. Model Validation and Performance Analysis

To investigate the capabilities of the new CCNet image processing and recognition model, a series of experiments were undertaken using PRPD images from the existing literature, where the types of defects had already been determined through manual PRPD pattern analysis. For these experiments, several different PRPD images representative of a wide range of defects were analyzed using the CCNet model. To avoid repetition, in the sections that follow, a small subset of these experiments is summarized to demonstrate how the effectiveness of the model was validated. While the capabilities for acquiring new PRPD images was available, a deliberate decision was made to use PRPD patterns that had already been classified by other researchers to ensure the impartiality of the evaluation of the new CCNet model’s capabilities. 

### 4.1. Experiment 1 (E1): Corona Discharges at Slot Exit

The authors of Reference [31] reported PRPD patterns obtained from discharges occurring in a 10 kV rated motor. The collected PRPD patterns demonstrated the monitoring of discharges across a range of voltage levels, spanning from 4 kV to 10 kV. Image pre-processing techniques, outlined previously, including noise reduction to eliminate less important regions and contrast and illumination enhancements, were applied to make the patterns more discernible.

The first experiment (E1) for testing the capabilities of the CCNet PRPD pattern recognition model involves corona discharge analysis where two PD measurements were performed at different voltages, 6 kV (Figure 3a) and 8 kV (Figure 3b), with the data presented in the form of PRPD plots. All PRPD patterns depicted exhibited positive polarity for both half cycles.

The cosine similarity scores presented in Table 2, which can range between 0 and 1, indicate how closely the PRPD patterns match the predefined templates based on the computation performed by the CCNet model. The PRPD patterns from slot discharges are expected to have a sharp-sloped triangular shape for both half cycles. As can be seen, for all patterns of E1, the CCNet algorithm consistently indicates a match with the “triangle” pattern with scores between 0.81 and 0.92. The second-best match is the “elevated arc” pattern with scores lower than 0.78.

Additionally, the mean score, *μ*, and standard deviation, *σ*, for each template have been evaluated. The mean score calculates the average cosine similarity between the PRPD patterns and each of the templates, where a higher mean indicates a closer match. As depicted in Figure 4, the highest mean score of 0.8427 corresponds to the “triangle” template. This signifies the greatest average resemblance of the PRPD patterns in E1 with the “triangle” shape pattern. The standard deviation quantifies the spread of the scores, with lower values denoting a tighter clustering around the mean and a higher consistency in the similarity of the PRPD patterns with each of the templates. While the standard deviation of the score for the “triangle” template is the highest, it is not significant enough to affect the result. At the same time, the substantially lower standard deviation scores for the other templates demonstrate the relative ‘confidence’ of the CCNet algorithm to produce a low match score, effectively confirming that the PRPD patterns in E1 do not match the other templates.

### 4.2. Experiment 2 (E2): Internal Slot Discharges

The second experiment, E2, extracts PRPD patterns that indicate internal discharges occurring within the slot of a stator winding, as reported in [31]. These discharges occur in the slot because of the loss of wedging pressure due to the settlement, chemical attack, abrasion, manufacturing deficiency, or erosion of the materials. PD measurements were conducted at various voltage levels, ranging from 4 kV to 10 kV, and the recorded patterns for 8 kV and 10 kV are shown in Figure 5. For reference, PRPD patterns corresponding to slot discharges are expected to exhibit a “baby stroller”-like shape. 

The cosine similarity scores for the slot discharge PRPD patterns compared to the predefined templates are shown in Table 3. The scores indicate a similarity to the “baby stroller” template, which matches the result of the manual pattern analysis pointing to slot discharges.

The high mean similarity scores (Figure 6) support the observation that the PRPD patterns are more similar to the “baby-stroller” template. The standard deviations corresponding to all templates are relatively low and close to each other in absolute terms, ranging from 0.0183 to 0.0310, again affirming the confidence in the identification of internal slot discharges.

### 4.3. Experiment 3 (E3): Void Discharges in a Rotating Machine

In the third experiment, E3, 2D PRPD patterns identified as void discharges resulting from air-filled pockets embedded in the main insulation of a rotating machine were obtained, as depicted in Figure 7. These discharges occur when air-filled voids in the insulation become sites of electrical activity. The voids, most of the time, are formed during the manufacturing process or because of the deterioration of the insulation material over time. PRPD patterns corresponding to void discharges are expected to have an “elevated arc” shape according to the literature. As can be seen, the PRPD image of Figure 7 contains bipolar patterns; hence, it was used to test the CCNet model’s capability to handle these PRPD representations. 

Table 4 shows the cosine similarity scores for the resemblance of the patterns in E3 with the templates of Table 1. The CCNet model reports the highest similarity of the patterns to the “elevated arc” template as expected. 

Figure 8 shows the mean similarity scores and standard deviations for the patterns of E3. It is still possible to make a determination that the patterns resemble the “elevated arc”, although on this occasion, the triangle pattern’s score is very close, and further analysis might be required to provide more confidence in the exact defect identification. 

### 4.4. PRPD Pattern Recognition Speed Investigation

The proposed CCNet can provide a result indicating the type of an incipient fault based on PRPD image analysis quickly and efficiently. Table 5 outlines the entire processing timeline, from PRPD image input to pre-processing to the final step of calculating cosine similarity scores. The pre-processing phase, which includes noise reduction, illumination enhancement, contrast enhancement, and the extraction of PRPD patterns from a provided 2D PRPD image, typically completes in just over three seconds, as can be observed from the timings shown in Table 5. Furthermore, the computation of cosine similarity, the concluding post-processing step, requires just a fraction of a second. The overall pipeline duration indicates a highly efficient process. To provide context regarding the computational resource requirements, the tests in this study were conducted on a cost-effective and widely available Windows PC equipped with an Intel core i3 CPU clocked at 3.40 GHz and 8.00 GB of main memory.

## 5. Discussion

As demonstrated, the CCNet image processing and recognition model can reliably identify the defect type from the provided PRPD images, consistently scoring the most likely match with 80% and above. It can handle both unipolar and bipolar PRPD representations and images with different pattern intensities and different colors. Additionally, the pattern recognition is not affected by other features that might be present in the image such as the power frequency sinewave, the plot grid lines, or the axis labels. 

It is important to highlight the similarities and differences of the CCNet model’s behavior compared to that of a human operator undertaking the task of analyzing PRPD patterns. A human operator can, in certain situations, have doubts regarding the resemblance of a PRPD pattern to a specific pattern template. It is not difficult to observe, for example, that the “baby-stroller” pattern can be seen as a “triangle” pattern with an added ‘tail’ component on the right-hand-side. Similarly, the “elevated arc” can be construed as a “triangle” with rounded edges. Depending on how pronounced these features are, the easier the determination will be. In this respect, the CCNet model behaves in a similar manner. The scores provided can often show a close resemblance of a PRPD pattern to more than one template. The difference, however, is that unlike a human operator’s final determination, which unavoidably includes a certain degree of subjectivity, the CCNet model assigns a specific score to this resemblance allowing for the decision-making to be more objective by quantifying the similarity of the PRPD patters to the predefined templates. 

While the initial validation of the proof of concept has produced positive results, it is worth mentioning that there are further aspects of PRPD analysis requiring additional investigation before the CCNet model can be widely deployed: The work presented in this article has only used five templates to verify the applicability of the model for PRPD analysis, but there are several others defined in the literature. Also, a PRPD pattern could correspond to a different defect depending on the type of insulation system being monitored. These should be relatively straightforward additions given the flexibility of the solution and the ease with which additional templates can be incorporated.There are situations where more than one insulation defect is present at the same time, which results in more than one PRPD pattern appearing on the same image. The capability of the model to distinguish between combinations of overlapping patterns is something that will need to be investigated.PRPD patterns can evolve over time either because more data are recorded or because the severity of a defect is worsening. How quickly, in terms of the pattern evolution, an accurate determination can be made is another aspect worth exploring.While the shape of a PRPD pattern can be used to identify its type, it is not sufficient to make a judgement regarding its severity. For this, information such as the amplitude and/or phase angle are required. It is envisioned that such information will be able to be extracted from PRPD images in addition to the patterns to fully automate PRPD analysis in the future.

## 6. Conclusions

The aim of the research presented in this paper was to develop and validate a quick and efficient method for recognizing, extracting, and processing PRPD patterns from two-dimensional plots to identify specific defect types affecting electrical equipment. The contributions can be summarized as follows:An effective way of segmenting and extracting PD patterns from images of PRPD plots has been introduced. The process involves delineating the region of interest and characteristics within the data, which are useful for PD analysis. This serves as the input to the subsequent image processing algorithm.A novel approach for PRPD pattern recognition has been developed that uses the cosine similarity function as the final step of an image processing pipeline. This allows for the patterns extracted from the images and processed by the newly developed model, CCNet, to be matched with predefined templates corresponding to known defect types.The effectiveness of the pattern recognition approach has been validated using several PRPD images reporting different types of defects. After processing, the type of defect reported by the CCNet model was compared to the reported defect type following manual PRPD analysis, and it was found that in all cases, there was a positive match.

The novel PRPD pattern recognition methodology that has been developed offers several advantages compared to other PD analysis techniques:**It is monitoring equipment-agnostic.** It is not tied to equipment supplied by a specific manufacturer and can be used with monitoring systems that are already in use. It can process any PRPD image regardless of how the measurements were taken and how they are presented as long as they are in the form of a PRPD plot.**It is fast and efficient.** Analyzing a PRPD pattern and reporting the similarity score indicating the type of defect takes only a few seconds. Furthermore, the computational resources required to deploy the CCNet model are minimal since it does not require training that relies on extensive datasets.**It is flexible.** The model can be adapted to include any number of templates of known defects and can be employed as narrowly or as widely as deemed necessary, for example to detect defects for a specific piece of equipment or for an entire facility with multiple different assets.

Future work will seek to fully test the capabilities of the newly developed method to encompass various electrical assets. Additionally, the pattern recognition capabilities of more obscure patterns will be investigated, and other aspects of performance will be finetuned.

## Figures and Tables

**Figure 1 sensors-24-03565-f001:**
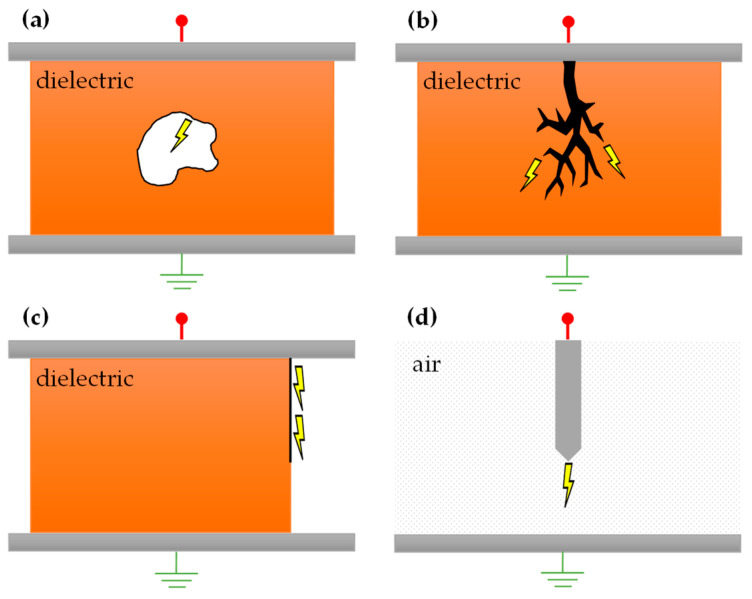
Examples of typical PD causes: (**a**) cavity without direct contact with electrode, (**b**) electrical treeing, (**c**) surface tracking, and (**d**) corona discharge.

**Figure 2 sensors-24-03565-f002:**
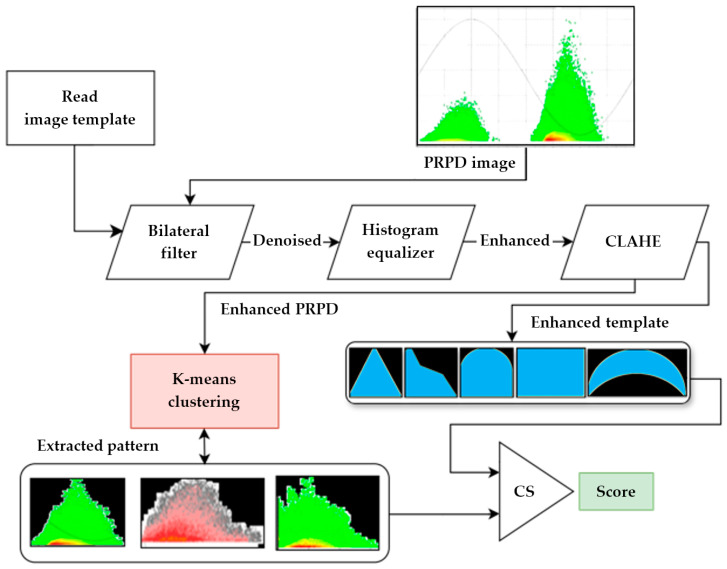
An illustration of the proposed CCNet algorithm for PRPD analysis, showing the information flow of the entire processing pipeline.

**Figure 3 sensors-24-03565-f003:**
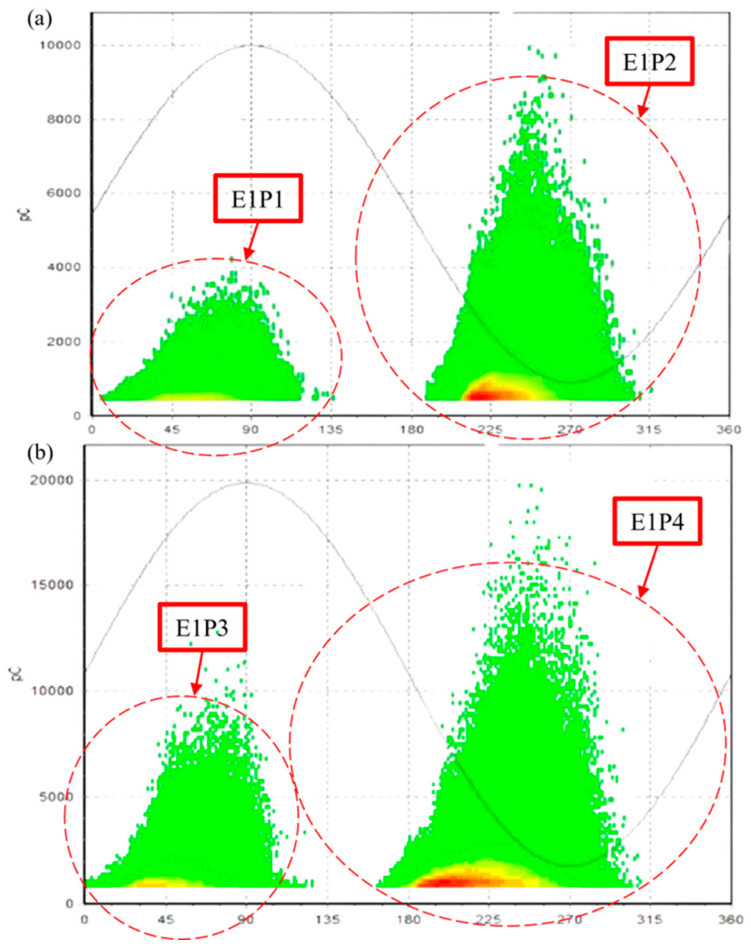
PRPD patterns of corona discharges at slot exit of motor: (**a**) corona PRPD image at 6 kV and (**b**) corona PRPD image at 8 kV.

**Figure 4 sensors-24-03565-f004:**
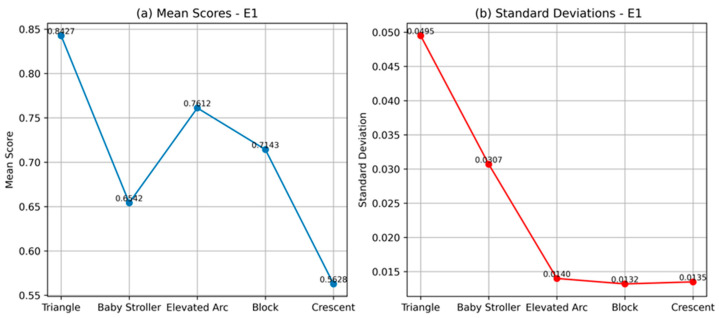
A comparison of E1 pattern recognition (**a**) mean scores showing corona discharges from a slot exit of a motor manifesting as triangular shapes and (**b**) standard deviations.

**Figure 5 sensors-24-03565-f005:**
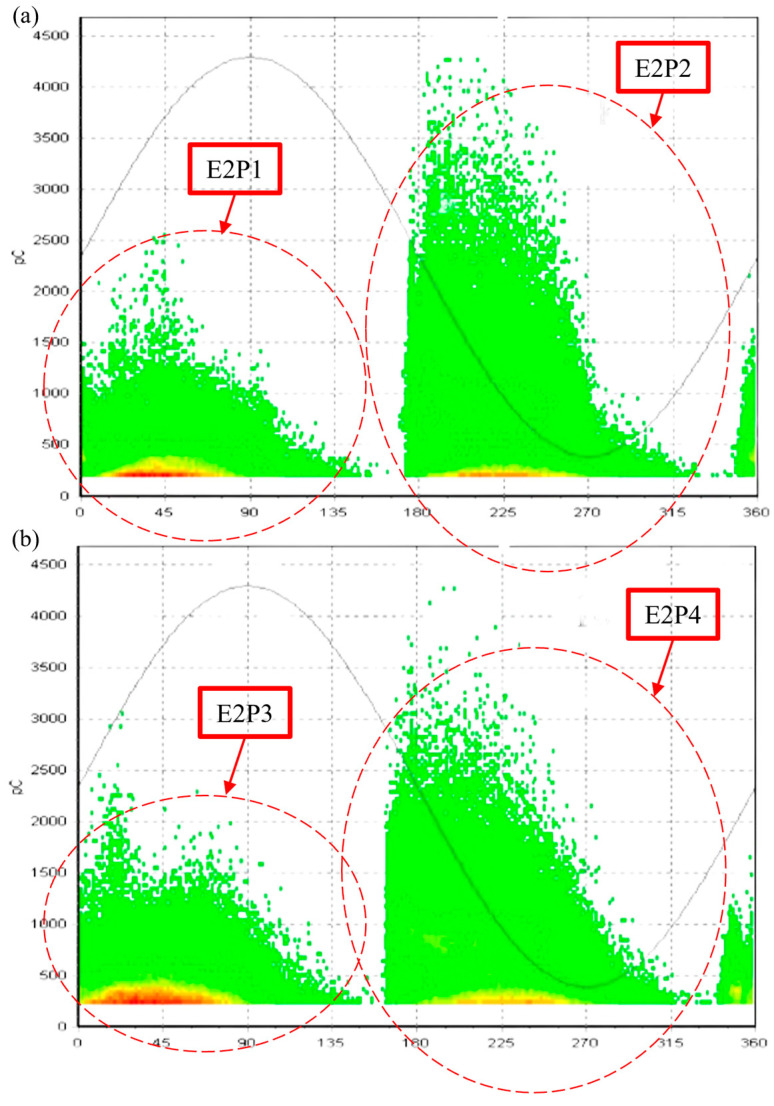
PRPD patterns of motor stator slot discharges (**a**) at 8 kV and (**b**) at 10 kV.

**Figure 6 sensors-24-03565-f006:**
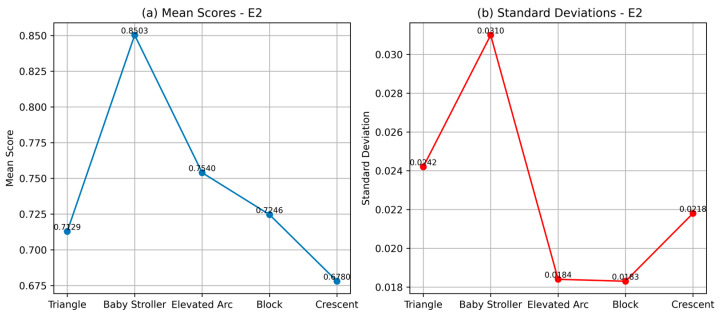
Comparison of E2 pattern recognition (**a**) mean scores showing slot discharges manifesting as baby stroller-like shapes and (**b**) standard deviations.

**Figure 7 sensors-24-03565-f007:**
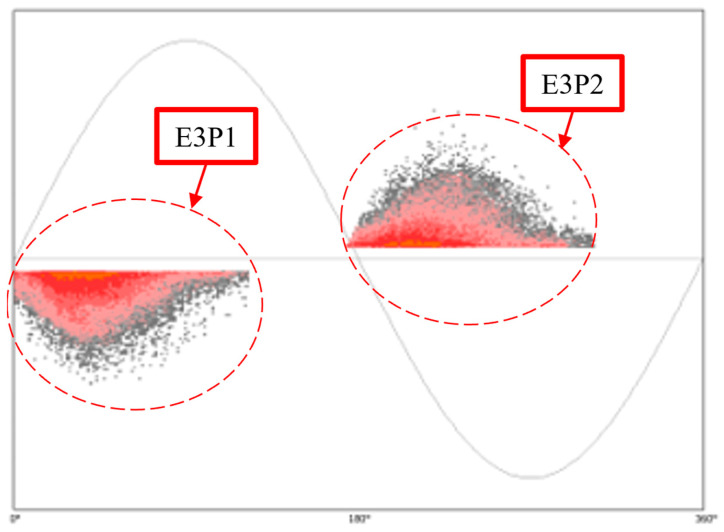
PRPD pattern of void discharges in a rotating machine.

**Figure 8 sensors-24-03565-f008:**
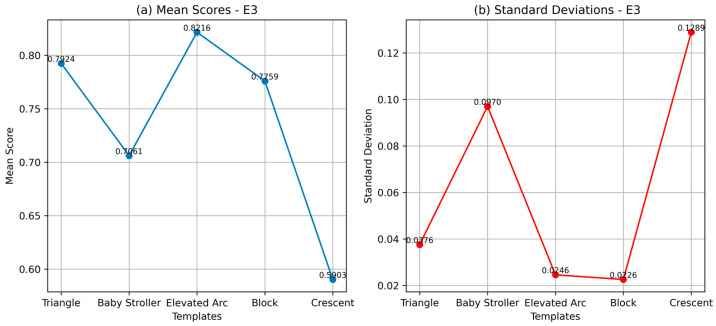
Comparison of E3 pattern recognition (**a**) mean scores showing slot discharges manifesting as elevated arc-like shapes and (**b**) standard deviations.

**Table 1 sensors-24-03565-t001:** Five typical PRPD patterns extracted from existing literature to be used as templates for pattern matching.

Pattern Shape	Pattern Name	Typical Defect Type
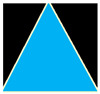	“triangle”	Corona discharges
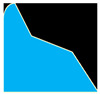	“baby stroller”	Slot discharges
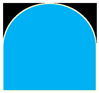	“elevated arc”	Void discharges
	“block”	Gap-type discharges
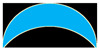	“crescent”	Surface discharges

**Table 2 sensors-24-03565-t002:** CCNet similarity scores for the PRPD patterns of E1 indicating the highest similarity with the “triangle” pattern.

PRPD Pattern	Template				
	Triangle	Baby Stroller	Elevated Arc	Block	Crescent
E1P1	0.81	0.65	0.78	0.73	0.56
E1P2	0.92	0.70	0.76	0.71	0.58
E1P3	0.82	0.64	0.75	0.70	0.57
E1P4	0.82	0.64	0.76	0.71	0.55

**Table 3 sensors-24-03565-t003:** CCNet similarity scores for the PRPD patterns of E2 indicating the highest similarity with the “baby-stroller” pattern.

PRPD Pattern	Template				
	Triangle	Baby Stroller	Elevated Arc	Block	Crescent
E2P1	0.72	0.86	0.77	0.74	0.67
E2P2	0.67	0.81	0.74	0.71	0.70
E2P3	0.73	0.88	0.74	0.71	0.65
E2P4	0.72	0.85	0.77	0.74	0.68

**Table 4 sensors-24-03565-t004:** CCNet similarity scores for the PRPD patterns of E3 indicating the highest similarity with the “elevated arc” pattern.

PRPD Pattern	Template				
	Triangle	Baby Stroller	Elevated Arc	Block	Crescent
E3P1	0.77	0.64	0.80	0.76	0.50
E3P2	0.82	0.77	0.84	0.79	0.68

**Table 5 sensors-24-03565-t005:** PRPD pattern processing and analysis speed using the CCNet model.

PRPD Image	Pre-Processing Time (s)	Post-Processing Time (s)	Overall Pipeline Duration (s)
E1 (a)	0.671	0.001	0.672
E1 (b)	1.005	0.001	1.006
E2 (a)	3.202	0.004	3.206
E2 (b)	3.200	0.005	3.205
E3	0.744	0.000	0.744

## Data Availability

The original contributions presented in this study are included in the article; further inquiries can be directed to the corresponding author.

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
