# Peer review of "Phase-Resolved Partial Discharge (PRPD) Pattern Recognition Using Image Processing Template Matching"

_sensors, 2024, doi:10.3390/s24113565_

Round 1

Reviewer 1 Report

Comments and Suggestions for Authors

The article discusses the issue of identifying defects in the insulation system based on analysis phase resolved Partial discharge patterns. The issue is certainly still relevant and diagnostic techniques based on partial discharge measurements are still being developed and are gaining in importance (especially in terms of online monitoring). The proposed analytical technique therefore fits into general development trends and has appropriate scientific potential for a journal of the rank of MDPI Sensors.

In general, it should be concluded that the article is very well prepared in terms of editing. It has appropriate graphics quality and a logical structure. The authors presented the issue and the proposed new algorithm in a detailed and very clear way. I found a few errors in the text structure in the work. I also have no objections to the choice of cited literature and the scope.

The following substantive reservations are only of a debatable nature and should be taken into account in the authors' further work. They do not affect the overall rating of the article, which is high.

1) In the presented method, the analysis is based on a graphic image generated by the measurement system. However, during measurements, the same data may be presented in different ways and at different scales. While conducting the analysis, the authors selected examples from the literature that met certain criteria. Unfortunately, there is no clear information on what PRPD images prepared for analysis should look like, whether they should be appropriately scaled, whether they should have the appropriate intensity, color, etc.

2) The analysis is based on patterns of selected defects, which are reduced to a simplified number of five defects. In the literature, you can find more extended standard databases, depending on the type of insulation system (it will differ for systems with gas, solid, liquid, laminated insulation, etc.). The algorithm can probably be adapted to individual variants, but therefore generalizations should be avoided when it comes to conclusions and the overall assessment of applicability.

3) The work assumes the existence of one defect, which unfortunately is not always the norm. The algorithm should be tested in the event of defects operating simultaneously, because this can only indicate its effectiveness and unambiguity. The results presented in the article already indicated, in some cases, the proximity of parameters that indicated two different defects, so it may turn out that the sensitivity of the method will be problematic. There are defects that can even cover images from other defects because they are simply more intense. This leads to the conclusion that the developed analytical method will have limitations, which are worth mentioning.

Editorial errors:
1) line 236: there is „while a lower score indicates greater similarity” should be „while a lower score indicates lower similarity”

2) Line 361: there is “recognistion” should be “recognition”

Author Response

Report on actions resulting from Reviewers comments

The authors are grateful to the reviewers for their comments, and for the opportunity to improve the paper.

Reviewer: 1

  1. In the presented method, the analysis is based on a graphic image generated by the measurement system. However, during measurements, the same data may be presented in different ways and at different scales. While conducting the analysis, the authors selected examples from the literature that met certain criteria. Unfortunately, there is no clear information on what PRPD images prepared for analysis should look like, whether they should be appropriately scaled, whether they should have the appropriate intensity, color, etc.

The reviewer’s comment is directly on point and in fact it is one of the main motivators for conducting this study in the first place. Unfortunately, there are no standards that define how a PRPD image should look like. As a result, PRPD images can vary in many ways such as scale, intensity, colour, etc. depending on the equipment manufacturer and consequently this is one of the factors that makes PRPD analysis difficult to implement on a wide scale. As mentioned in the introduction: “The approach is designed to be independent of the specifics of PD measurement techniques and the visual representation of PRPD patterns”. This is also emphasised in the conclusions. In other words, the method introduced in the paper can handle any PRPD image, irrespective of scale, intensity, or colour, as long as it was created using the f-q-n representation and the partial discharges are synchronised with the power cycle. One of the reasons we have chose to use images from existing literature rather that producing our own is exactly this, so we could evaluate the capabilities of the approach to handle a wide range of different PRPD image representations. In the paper itself we do include examples of PRPD images that look substantially different in terms of colour, scale, intensity e.g. figure 5 from figure 7. Other examples have also been used for validation that are not included in the paper to avoid repetition.  

  1. The analysis is based on patterns of selected defects, which are reduced to a simplified number of five defects. In the literature, you can find more extended standard databases, depending on the type of insulation system (it will differ for systems with gas, solid, liquid, laminated insulation, etc.). The algorithm can probably be adapted to individual variants, but therefore generalizations should be avoided when it comes to conclusions and the overall assessment of applicability.

The reviewer’s comment is correct. There are more than five PRPD patterns documented in literature and in fact even in the references that we have provided there are more than five available. The work detailed in the paper is not meant to describe the exhaustive and fully implemented development of a commercially applicable solution. It is a proof-of-concept demonstration that explains how the concept works and how it could potentially be used if fully developed. PRPD patterns can have different meanings i.e. point to a different defect based on which asset is being monitored. They could also change over time or be influenced by the sensor being used to capture the raw measurements. We are aware of these and other aspects that can influence PRPD analysis and we are planning to investigate them in the future to make the proposed method as robust as possible. Nevertheless, one of the main benefits of the proposed method, as mentioned in the paper, is its flexibility. Pattern templates can be incorporated into the software in a matter of minutes. Also, it would be extremely easy to configure the software to only include templates corresponding to specific insulation systems or assets. We have been careful not to suggest that the method is immediately applicable in its current state. We have merely suggested that the results, which objectively are positive, “indicate potential” for future applications.

  1. The work assumes the existence of one defect, which unfortunately is not always the norm. The algorithm should be tested in the event of defects operating simultaneously, because this can only indicate its effectiveness and unambiguity. The results presented in the article already indicated, in some cases, the proximity of parameters that indicated two different defects, so it may turn out that the sensitivity of the method will be problematic. There are defects that can even cover images from other defects because they are simply more intense. This leads to the conclusion that the developed analytical method will have limitations, which are worth mentioning.

The reviewer raises another valid point. In many situations there could be more than one defect present, and this could change the observed patterns. This is one of the aspects that we intend to investigate in the future. There is also the possibility that more intense patterns can obscure less intense patterns, but this is an inherent problem with PRPD detection which would also be faced by a human operator.  As mentioned previously, the study is a proof-of-concept demonstration and there are aspects that will require further investigation. The aspect of sensitivity is an important one and it is discussed in the “Discussion” section of the paper. The reporting of similarity scores is open and transparent. It provides a quantifiable measure of resemblance of a PRPD pattern to all predefined templates. The method is intended to be used as an additional tool to aid decision-making not to take decision-making away. The ultimate decision of whether this measure is acceptable to make a definitive call is still up to the asset manager/operator. The only way to gain confidence in the capabilities of this or any other automation method so it can be used without human supervision would be to extensively test it in the field, but this goes beyond the scope of this study.

In response to the reviewer’s comments, we have made additions to the discussion to include aspects discussed in the responses above.

Reviewer 2 Report

Comments and Suggestions for Authors

The partial discharge is one of important indexes representing the insulation status of high voltage electrical equipments, such as GIS, GIL and transformer. The automatic recognition methods of partial discharge patterns attract extensive attention and have been studied intensively, in which deep learning algorithms were utilized in general.  This paper proposed a novel PRPD pattern recognition method, CCNet, which uses the Cosine Similarity socres of 2D plots to implement image processing template matching. The authors also provide several typical experimental discharge images to validate the algorithm and demonstrate its computation time.  Therefore it is a quick and efficient method for recognizing PRPD patterns.

Suggest that the authors could check the equations in Section 2 ~ Section 3, to avoid using the same symbol or easily-confused symbols, 'a' and '𝛼' (alpha) for different variables. In addtion, how to project a 2D vector of PD image to 1D vector should be clarified in the text, as the variables of a and b are 1D vectors in equations (6), (7).

Author Response

Report on actions resulting from Reviewers comments

The authors are grateful to the reviewers for their comments, and for the opportunity to improve the paper.

Reviewer: 2

  1. Suggest that the authors could check the equations in Section 2 ~ Section 3, to avoid using the same symbol or easily confused symbols, 'a' and '?' (alpha) for different variables.

We appreciate bringing up this significant point. It has been duly resolved as outlined in the updated manuscript. Furthermore, we have taken care to define each parameter variable consistently throughout different sections, ensuring clarity for readers to understand its meaning in each specific context without ambiguity.  

  1. how to project a 2D vector of PD image to 1D vector should be clarified in the text, as the variables of a and b are 1D vectors in equations (6), (7).

We express our sincere gratitude to the reviewer for bringing this crucial point to our attention. In response, we have clearly documented in the revised manuscript the utilization of a flattening function to transform the 2D vectors.